# Experiences of Primary Healthcare Workers in Australia towards Women and Girls Living with Female Genital Mutilation/Cutting (FGM/C): A Qualitative Study

**DOI:** 10.3390/healthcare11050702

**Published:** 2023-02-27

**Authors:** Olayide Ogunsiji, Anita Eseosa Ogbeide, Jane Ussher

**Affiliations:** 1School of Nursing and Midwifery, Liverpool Campus, Western Sydney University, Locked Bag 1797, Penrith, NSW 2751, Australia; 2School of Health Sciences, Campbelltown Campus, Western Sydney University, Locked Bag 1797, Penrith, NSW 2751, Australia; 3Translational Health Research Institute (THRI), School of Medicine, Western Sydney University, Penrith, NSW 2751, Australia

**Keywords:** female genital mutilation/cutting, primary healthcare providers, experiences, education and training, Australian healthcare

## Abstract

Female genital mutilation/cutting (FGM/C) is a harmful cultural practice with significant health consequences for affected women and girls. Due to migration and human mobility, an increasing number of women with FGM/C are presenting to healthcare facilities of western countries (including Australia) where the practice is non-prevalent. Despite this increase in presentation, the experiences of primary healthcare providers in Australia engaging and caring for women/girls with FGM/C are yet to be explored. The aim of this research was to report on the Australian primary healthcare providers’ experiences of caring for women living with FGM/C. A qualitative interpretative phenomenological approach was utilised and convenience sampling was used to recruit 19 participants. Australian primary healthcare providers were engaged in face-to-face or telephone interviews, which were transcribed verbatim and thematically analysed. Three major themes emerged, which were: exploring knowledge of FGM/C and training needs, understanding participants’ experience of caring for women living with FGM/C, and mapping the best practice in working with women. The study shows that primary healthcare professionals had basic knowledge of FGM/C with little or no experience with the management, support, and care of affected women in Australia. This impacted their attitude and confidence to promote, protect, and restore the target population’s overall FGM/C-related health and wellbeing issues. Hence, this study highlights the importance of primary healthcare practitioners being skilled and well-equipped with information and knowledge to care for girls and women living with FGM/C in Australia.

## 1. Introduction

Over centuries, humans have developed a range of practices that are tightly tied to complicated social structures and conventions [1], with some of these practices violating human rights [2]. Female genital mutilation/cutting (FGM/C) is one of the practices known to impose lifelong effects on the health and wellbeing of women and girls in developing countries [3]. However, due to increased global migration, FGM/C has become a global issue [4]. This human movement has allowed for countries to work together to protect girls from being circumcised and provide the best possible healthcare for women and girls who have undergone FGM/C [5]. According to the World Health Organization (WHO), over 200 million girls and women in 29 African and Middle Eastern countries (e.g., Yemen, Iraq) have undergone FGM/C, with three million more at risk each year [6] It is also prominent in a number for Asian countries (for example, India, Malaysia, and Indonesia) as well as among migrant populations in Australia, New Zealand, the United States, and Europe [7]. If nothing is done about the current situation, it is estimated that approximately 4.1 million girls will be at risk of being cut by 2030.

FGM/C is characterised as any procedure involving the partial or total removal of the external female genitalia or other injury to the female genital organs, whether for cultural, religious, or other non-therapeutic reasons [8]. This procedure has no medical benefit or medical explanation [9] and is often carried out on young girls between infancy and age 15 [6]. This practice can disrupt normal bodily systems and cause lasting physical and sexual health issues. It is also known to have serious and long-term psychological repercussions, such as anxiety, depression, and post-traumatic stress disorder (PTSD) [10]. Pregnancy and childbirth can be a particularly difficult period for women who have undergone FGM/C, as the physical and psychological difficulties of FGM/C may be aggravated [11,12,13] and can also cause sexual disability [14]. These issues cause women and girls to seek medical attention, yet data reveals that health systems in western countries are unable to respond owing to a lack of infrastructure or competence [15]. 

### 1.1. Female Genital Mutilation/Cutting: Australian Context

The prevalence of female genital mutilation among Australian women is unknown [5]. However, it has been estimated by the Australian Institute of Health and Welfare [16] that approximately 53,000 females born outside of Australia but residing in Australia have undergone FGM/C (i.e., 4.3 females per 1000 females in Australia). As a result, it is important to prioritise vulnerable populations and people who have been impacted by FGM/C, individually or collectively, in order to address their physical, psychological, and human rights needs [17]. 

In light of this estimation of human rights issues and health impact, Australia and New Zealand are among the 12 developed nations to pass anti-FGM/C legislation outlawing all forms of genital mutilation [18]. The practice of FGM/C is outlawed in all Australian states and territories. This includes sending a person abroad to have a procedure performed, as well as facilitating, supporting, or encouraging others to do so. By extension, the Australian government firmly believes that tradition or culture cannot be used to justify the violation of the human rights of women and girls [19]. The Australian legal and policy prohibition on FGM (including “medicalised” FGM by nicking) appears to be warranted based on the health effects, breaches to autonomy and liberty, and lack of compelling justification [18]. This is a progressive step in the prevention and protection of girls and women living in Australia [17]. To further promote cultural safety in the healthcare system for women and girls in Australia who have experienced FGM/C, the Multicultural Centre for Women’s Health created the National Education Toolkit for FGM/C Awareness (NETFA), a best-practice guide that serves as a national standard for culturally competent FGM/C health promotion initiatives [20]. Additionally, the health and community sectors are committed to advancing the understanding and management of FGM/C. The Australian Medical Association and the Royal Australia and New Zealand College of Obstetricians and Gynaecologists (RANZCOG), for example, have produced position statements on FGM/C with a clear stance urging all health practitioners to refuse requests for elective reinfibulation or other forms of FGM since they confer no health benefits to women and, on the contrary, present a vast array of negative health implications [21]. The Australian College of Nursing and Australian College of Midwives have established an online learning centre on FGM/C [22,23,24]. These examples show that the Australian healthcare system is moving in the right direction. 

With the progress being noted, evidence has shown that there is still room for improvement in developing an awareness of FGM/C among healthcare professionals [25]. Turkmani et al. [26] indicated in their study that many midwives were unfamiliar with the law or health data pertaining to FGM/C, and they were unaware of referral pathways for affected women in Australia. Similarly, a study by Dawson et al. [27] highlighted that many Australian midwives lacked confidence and competence when it came to caring for women who had undergone FGM/C. Comprehensively, Evans et al. [28] captured the shortcomings of healthcare professionals (particularly nurses and midwives) in providing care for FGM/C-affected women and girls in Australia. They reported that there is insufficient knowledge, clinical skills, cultural competency, confidence, and awareness of legal duties to successfully address the needs of women with FGM/C, all which are necessary in the provision of woman-centred care. 

Hence, it is evident that empirical research on FGM/C in Australia has focused on midwives and nurses. While this is an important health service delivery cohort, it is also significant to capture primary healthcare workers as a cohort. These healthcare workers, inclusive of nurses, general practitioners (GP), pharmacists, and allied healthcare providers, are most likely to be the first point of contact for women and girls in their immediate community [29]. These target populations are important to explore because the impact of FGM/C goes beyond physical health and also includes psychological, social, sexual health, and wellbeing [30]. Therefore, the aim of this research was to explore the experiences of Australian primary healthcare providers towards women living with FGM/C. These reported experiences will inform the development of strategies for capacity building of primary healthcare providers in the area of FGM/C, which will answer our research question stated below:

### 1.2. Research Question

What are the experiences of Australian primary healthcare providers in caring for women living with FGM/C?

### 1.3. Terminology

In line with many international organisations, ‘female genital mutilation/cutting’ or ‘FGM/C’ reflects the importance of using non-judgemental language that is respectful of individuals who have undergone the practice. Inclusion of the term ‘cutting’ is not an attempt to excuse or diminish the impact of the practice, but to acknowledge the different ways girls and women might identify or interpret their experience. Particularly in countries of destination such as Australia, best practice health promotion and community development programs agree that using appropriate language to ensure that communities are not marginalised or stigmatised is more effective in engaging communities and facilitating dialogue [31].

## 2. Materials and Methods

### 2.1. Research Design

An interpretative phenomenological analysis (IPA) informed the study, which is important to understand the phenomenon from the viewpoints of the participants [32]. It emphasises what a lived experience means to the individual through a process of in-depth reflective investigation [33]. Through a process of in-depth reflective enquiry, IPA seeks to discover what an individual’s lived experience means to them. IPA employs phenomenological reasoning in order to return ‘to the phenomena themselves’ [33]. However, IPA also recognises that we are influenced by the environments we inhabit and the experiences we encounter. Therefore, IPA is a process of interpretation between the researcher and the researched, primarily influenced by Heidegger’s interpretive phenomenology, hermeneutics, and idiography. IPA provides recommendations on how to approach a phenomenon of interest, including sampling, data collecting, and analysis [34].

### 2.2. Positionality Statement

Reflexivity and fairness ensured this study’s trustworthiness, including reflection on how one’s history, experiences, values, and identity shape one’s research [32]. Authors O.O. and A.O. are from one of the African countries where type II FGM/C is currently 60 percent prevalent. Author J.U. is a woman from a Western country with substantial experience in women’s reproductive health research. These were taken into consideration when planning the study, conducting the interviews, and evaluating the data. 

### 2.3. Recruitment & Sample

Convenience sampling was employed to recruit participants from primary healthcare providers from primary healthcare settings. The participants in the study were drawn from GP clinics, community health centres, and women’s health centres in Blacktown, Fairfield, Liverpool, Parramatta, and Bankstown. These five suburbs have significant populations from countries where FGM is prevalent [35] and are therefore likely to present to primary healthcare professionals for healthcare there. The GP clinics were accessed through the Western and South Western Sydney Primary Health Networks that cover the locations of interest in this study. Further, a flier that contained the details of the study and a Participant Information Sheet was attached to the email sent to the stated healthcare settings, and the Chief Executive Officers (CEOs) and managers were requested to kindly post the flier on their notice boards and websites and distribute the information to their networks. Participants who were interested in taking part in the study contacted the first author who provided additional information and clarification required by the potential participants.

To participate in the study, participants needed to satisfy the following inclusion criteria: (1) working in a primary healthcare setting such as a GP clinic, women’s health centre, or community health centre in Blacktown, Fairfield, Liverpool, Parramatta, or Bankstown; (2) be 18 years or over; and (3) willing to participate in the study. To be included in this study, participants needed to be part of the team involved in the provision of care for women and girls that received care in women’s health centres. The holistic approach to healthcare in women’s health centres requires that the team has to be part of the primary healthcare giving, including the administrator who is responsible for the client’s intake at reception as well as those involved in court advocacy. We included administrators and legal assistants. This is because women’s experiences with these administrators and legal assistants could have a negative influence on them accessing the women’s health centre, in terms of engagement with the service. To further recruit more participants in the study, the snowballing technique was also used to enable participating professionals to introduce the study to their networks, which allowed for a total of 19 participants to participate who represented the 5 suburbs from the different healthcare settings. Recruitment saturation was reached based on the number of participants who came forward and indicated interest to participate in the study.

### 2.4. Data Collection

Data was collected between January and July 2019. The primary healthcare professionals were engaged face-to-face or by telephone in semi-structured, in-depth, audio-taped interviews lasting approximately 40–60 min. The interviews were conducted in quiet rooms or offices of primary healthcare centres where 17 of the participants were working. All interviews were conducted by author O.O. This ensured consistency in the interview approach and process. Prior to the commencement of the interviews, all of the participants were given a participant information sheet outlining the details of the research. They were also given the opportunity to ask questions and were reminded that their participation in the study was voluntary. All of the participants who took part in the study provided written consent and their names were redacted with pseudonyms assigned prior to data analysis. All of the participants responded to an initial question “Can you please tell me what you know about Female Genital Mutilation?” Subsequent questions depended on the participants’ responses to the initial question. Probes and paraphrasing were used during data collection to ensure that participants provided adequate insight into the questions they were asked. A compilation of the questions to which the participants responded is presented below in the interview schedule.

### 2.5. Interview Schedule

Can you please tell me what you know about Female Genital Mutilation (FGM/)?Please mention the different types of FGM that you know and can you please describe each of them?How knowledgeable do you consider yourself in the identification of the various types of FGM?What are your experiences in caring for people living with FGM?Can you please tell me your specific concerns about caring for people living with FGM?Are there any areas of training needs that you think of in caring for people living with FGM?Are there any other things you would like to talk about in relation to your knowledge, experience, and training needs in caring for people living with FGM?

### 2.6. Data Analysis

During the data collecting phase, qualitative data analysis begins when the researcher confirms that the data acquired feeds and informs the following data collected [32]. Interview information was transcribed verbatim and manually analysed. Yin [36] proposed a five-phased cycle for qualitative data analysis, which further led the analysis. These steps include accumulation, disassembling, reassembling, interpretation, and conclusion. The accumulation process involved organising the data in an order. During this phase, all 19 transcripts were reviewed for accuracy and completeness by listening to the audio recordings; the fieldnotes were expanded into more detailed notes; and a word document folder including the transcripts and field notes was created and stored. This was part of the first phase of being acquainted with the stories of the participants. 

The disassembling phase concentrated on fragmenting the compiled data into smaller parts [36]. Both authors (O.O. & J.U.) with substantial expertise in qualitative research analysis read the transcribed material hermeneutically as a back-and-forth process between the parts and the whole [37]. The demographic information was summarised, and recurring words, phrases, and sentences were detected. These recurring sentences, phrases, and words were investigated in depth during the assembling phase, which involved a substantial cluster of codes or motifs. The responses of the participants were meticulously organised in accordance with the study’s purpose and research questions. The individuals’ responses to identical questions were consistently compared and questioned.

In the interpretive phase, “reassembled material is used to generate a new story” [36] (p. 179). In this study, the researchers employed a combination of demographic data analysis and participant replies to convey a more complete tale. Several themes were extracted from the interview data before a consensus of themes was reached. The concluding phase derived inferences from the complete dataset. The quotations were carefully reviewed to ensure that the meanings were kept in the form in which they were provided by the respondents. Participants names were redacted and pseudonyms were assigned.

### 2.7. Ethics Approval

This study was approved by Western Sydney University Human Research Ethics Committee (HREA H13021).

## 3. Result

The study included 19 primary healthcare workers from women’s health centres, GP practices, and community clinics, as well as allied healthcare workers and triage employees. The participants in this study included a GP (n = 1), women’s health nurses (n = 3), registered nurses (n = 3), a relationship educator (n = 1), women’s health centre administrative staff (n = 2), counsellors (n = 4), a program coordinator (n = 1), social workers (n = 2), a massage therapist (n = 1), and a legal assistant (n = 1). They were all female, aged 20–70 years. People from Australia, Africa, the UK, Fiji, and New Zealand made up the majority (n = 11). Most (n = 14) had never looked after an FGM/C-affected woman or girl, whereas the others had over 5 years’ experience. The study sample had varied health and social workers who may have interacted with women or girls who had experienced FGM/C. There is a need for every tier of the health and social sector to be up-to-date in providing culturally competent and safe healthcare that empowers these women/girls to feel in control of their health decision and fosters trust and confidence that allows for their voices to influence community outreach programs, speak up legally to persecute perpetrators of FGM/C, and enhance their health-seeking behaviours to promote their physical, social, religious, and mental wellbeing. Three themes and three subthemes emerged from the data analysis exploring the experiences of caring for women living with FGM/C. The major themes identified included exploring knowledge of FGM and training needs, understanding participants’ experience in caring for women living with FGM, and mapping the best practice in working with women (see Table 1).

### 3.1. Exploring Knowledge of FGM and Training Needs

“I think my knowledge is quite limited regarding FGM”: Having a lack or patchy knowledge of FGM/C.

The participants’ knowledge of FGM varied in depth, “To begin with, I think my knowledge is quite limited regarding female genital mutilation” (Angela) and “…I only know little bit of information about female genital mutilation” (Alex). Drawing from their tone of voice and body language, only 15 participants across all the professions in this study confidently spoke about their understanding, such as: “I know that FGM is also referred to as female circumcision, it’s also referred as cutting but FGM is the common name used to for it” (Sam) (nodding her head in confirmation). This knowledge was mainly centred around where FGM/C is practiced, Australian’s stance on FGM/C, reasons behind the practice, other names for FGM/C, typologies, age at which it is practiced, and the parts of the woman’s body that are affected. 

Out of the 15 participants who boldly expressed their understanding of FGM/C, only one women’s health nurse knew much more than the others. This was in terms of her knowledge of other names for FGM/C, typologies, where it is practiced, Australia’s stance on the practice, and current prosecution in Australia. Meanwhile, she did not talk about the international attention drawn to the practice.

Among the remaining participants who were not confident in presenting their understanding of FGM/C, their understanding was limited to several areas about FGM/C. “Not much really, but basically yeah, that they get cut in the genital area, in the area of the clitoris and I don’t know if they sew them as well, I don’t know, and then it draws together and I don’t know the issue of marriage thing that they have, when they first have their sexual intercourse they’ll be opened up. Yeah that’s all I know” (Karen).

A few of the participants stated that they were occupying new roles with the need to care for populations from backgrounds where FGM/C is prevalent. They identified limited expertise in addressing the psychosocial issues of women living with FGM/C. They acknowledged that there is a high potential for women and girls to present with FGM/C-related health issues due to the growing number of migrants and refugees from backgrounds where the practice is prevalent. However, this was a far cry from having the capacity to address these psychosocial issues. They further expressed a lack of knowledge about what to ask the women, the required information to provide them, as well as signs and symptoms to look for.

The participants accounted for the sources of their incomplete and patchy knowledge of FGM/C. These included training/workshops, in the course of working, television documentaries, books, and from being born in countries where the practice is prevalent. The majority referred to some books they had read, and one person was specific that she read about Dr. Hamlin in Ethiopia and fistula.

The participants’ stories revealed that the knowledge gained from these sources were dated and incomplete, such as: *“*I watched some sort of documentary... I think it was on Foxtel, I think that is even ten years ago” (Heidi). Some of their training dated back 20 years with the most recent being three years ago at the time of this data collection: “I did the training aah far back…gosh…. that was maybe 20 years ago” (Joey). The duration of most of these training/workshops was one day and they described the content covered as brief and basic: “And so, they talked a little bit about it (FGM/C), it was actually a brief one-day workshop. They talked about, sort of the different types of circumcision, they talked about the impacts it could have and I guess how it was sort of socially acceptable within certain cultures and I guess why it could be done, yeah, it was very basic” (Samantha).

“I just found it extremely cruel…”: Having a negative attitude towards FGM/C.

It was interesting to note the difference in the attitude of the majority of the participants who knew little about FGM when compared with the few who were more confident about what they knew. The immediate emotion of the majority of participants was that FGM/C is bad and barbaric, drawing from television episodes, movies, documentaries, and dated information: “…the first word that just came to my head, just barbaric to do something like that” (Heide). They described the practice as terrible, traumatising to watch, shocking, and disturbing to see.

None of the participants described watching infants undergoing FGM/C only young girls of about nine years old (and sometimes older) and of African descent. Identified venues of the procedures included inside rooms, the bush, or places surrounded by trees. According to the participants, what they saw and heard in movies was screaming and characters being dragged away by elders. One participant stated that she the girl’s labia together stitched together with just a small outlet to allow urine and blood during her period. They expressed seeing the characters’ pain and shock and found this upsetting and mentioned that some of them changed the channel they were watching: “It was very upsetting; I changed the channel. I sort of got quite upset about it. What was upsetting was the pain that they went through, I think the girl that I was watching, she didn’t want it done. It was against her will. I found it very confronting” (Tamara). 

Some of the participants, including Victoria who was born in one of the countries where the practice is prevalent, condemned the practice as unjustifiable. Referring to their spirituality, one participant questioned why a woman’s created body parts would be tampered with, stating: “Why would we make difficult what God, or the creator created this part of the body, or the part of it, why would we intervene to do to make it something, to make it difficult for a child to be born?” (Mickey).

Relative to the negative attitude, some participants explained FGM/C as more of a cultural norm. Perhaps due to their FGM/C training or more than five years’ experience of caring for women living with FGM/C, this cohort of participants explained that they were aware that some healthcare providers refer to the practice as violence. They argued that this is due to a lack of understanding of the whole issue, which includes cultural expectations. “…it’s more of a cultural norm for a lot of women. I know that a lot of nurses think and see it as violence and sometimes I think yes…there’s an aspect of that but also, it’s the women’s culture and that’s part of their being a female in that culture. I don’t think people really understand the whole issue” (Joey).

“I just think we need training”: Need for training. 

There was an overwhelming demand for FGM/C training from all of the participants who took part in this study: “…just a starter is training, that’s it I mean 100% I just think we need training” (Sam). The majority of the participants stated that FGM/C was not part of their curriculum in the course of their formal education: “So certainly not anything about FGM/C that was addressed at university at all at any stage” (Julie). They provided some assumptions that they might not have been taught about FGM/C because the curriculum developers may have lacked an understanding of the prevalence rate or perhaps it was considered a medical issue. They highlighted that even though every university tends to have its own program, every University has some units that cover cultural diversity and cultural sensitivity. “Look, it would definitely be useful to embed it as part of you know, I mean each of the universities is different in terms of what kind of programs they have. Obviously, all of them cover issues around cultural diversity, cultural sensitivity but I would probably be surprised to find if any of them addresses this specifically at all within a social science degree” (Julie).

Some of the registered nurses who had previously worked in acute care settings, such as the emergency ward, stated that they had never come across any information about FGM/C. Surprisingly, even for community health nurses, the story was the same in terms of lack of information about FGM/C. Some of these community health nurses stressed that they go to people’s houses to check babies using a checklist. However, they stated that they had not come across a checklist that included female circumcision. They emphasised the need to train and create awareness not only among community nurses but also among child and family health nurses.

At the very least, the participants were eager to know the basics about FGM/C, such as the different types, what it is, how it is done, medical consequences, and psychosocial issues. They emphasised the need to know the referral pathway: “we would like to refer someone on but because it’s so under the radar you don’t know where to refer a person” (Karen).

### 3.2. Understanding Participants’ Experience of Caring for Women Living with FGM/C

Only five participants had the experience of caring for women living with FGM/C: “Yes, I have (cared for circumcised women), a couple of times (Ruby); “I have seen a number of women who have experienced, aah FGM” (Joey). The primary healthcare workers cared for the women in their clinics with presentations that included physical concerns about not being sewn up after childbirth and other issues of damaged female genitalia. Given that vaginal examination is a significant part of daily work routines for women’s health nurses, they were able to describe the physical damage to the female genitalia of the women they cared for. They spoke about caring for women who did not have a clitoral hood, those that were pierced through the clitoris as young girls, and more. “As a women’s health nurse, I saw a couple …there was the one that didn’t have the clitoral hood. And another one, I don’t know how to describe it, she had been pierced through the clitoris as a young girl” (Sarah).

However, some of the participants considered FGM/C as a special type of women’s health issue that they found challenging in terms of care provision. The only general practitioner in the study expressed her wealth of experience in the provision of female sexual and reproductive health. She explained that she had looked after women with a diverse range of random and extensive health issue. 

One major experience reported about caring for women living with FGM/C was the women’s lack of willingness to disclose information about their FGM/C. The specific contexts through which the women could make the disclosure included general conversations about abuse and specifically, sexual abuse. The participants cited instances where the women mentioned their experience of FGM/C but did not want to talk about it or do anything about it. Ruby identified specific examples of some women in domestic violence situations who disclosed FGM/C. She explained that she wanted to probe further, elicit the women’s feelings, medical treatment needs, and even organise a referral; however, the women refused to talk about it: “Both women in particular didn’t really want to talk about it. It was just something that happened. It was mentioned and I was exploring other topics with her. Both women were in a domestic violence situation and I was exploring that, and they mentioned that (FGM/C) but they didn’t want to talk further about that, and they didn’t want to do anything about that. That wasn’t their focus” (Ruby).

There were suggestions that the women’s hesitance to disclose detailed information about their FGM could/C be due to lack of trust in healthcare providers, fear of mandatory reporting, and thought of helplessness. “…there could be some distrust with people from ah… workers, people from professional backgrounds (Alex); “…fear among women that we may be…file a report or do something for them or judge them” (Joey); and “I guess the thought was that it is done, it’s too late. I just have to put up with this now” (Ruby).

Of particular concern to the women’s health nurses was that the women were not presenting for maternity services and cervical cancer screening. They suggested that cervical cancer screening may be a foreign concept for women from culturally and linguistically diverse background and further expressed the probability that the women might be seeing counsellors. However, the core concern was that this lack of presentation may put the women at risk of missing important women’s health services: “…because unless the women come to us, I still feel that, you know, they might be slipped, you know slip through the cracks” (Sarah). 

### 3.3. Mapping the Best Practice in Working with Women

The participants shared their thoughts on best practice for engaging and caring for women living with FGM/C. Irrespective of having the experience of caring for the women or not, they spoke about the importance of rapport, trust-building, and genuine interest in knowing the women and girls being cared for. They described how they navigate the sensitive conversation around FGM/C. According to the participants, their conversations did not commence with questions about FGM/C. Rather, they started by focusing on building rapport and getting to know the women more, such as their countries of origin and cultural beliefs. It is after this that they asked the women if they had undergone FGM/C and then proceeded to examine then. “…it (FGM/C) wouldn’t be the first question that I would ask… I think that with me 90% of the consultation time and building trust with someone and then I might have a sense that this woman is from a specific country of origin and that might come up around asking her, is that something that is prevalent in your community, is that something that she has experienced herself… before I even begin to examine her” (Joey).

Depending on their professional background, the participants explained that they knew the angle through which to approach their engagement with the women. For example, Ruby explained that as a counsellor, she would focus on the emotional impact of FGM/C on the lives of affected women and girls in the first instance, then inquire about other health issues and make a referral as required, stating: “I will be looking at the emotional impact to the girl and how it has affected the whole life. That’s what I will be addressing first and then I will be asking her whether there is any physical impact, whether she is suffering anywhere that is causing any problems and I will be referring her. We have here in the Centre, a doctor and I will be doing a referral to the doctor to determine if there is anything going on there” (Ruby).

Not surprisingly, a few experienced clinicians drew on their knowledge of anatomy and physiology of female genitalia. They emphasised their ability to identify FGM/C and preparedness to care for the women as best practice: “I was prepared to care for those people” (Sarah). They emphasised that, at the very least, healthcare providers need to know about available services that can support people who have experienced FGM/C, to provide required information, and refer them accordingly.

Some of the participants that did not have the experience of caring for women living with FGM/C (but had worked for several years in the general area of women’s health) further spoke about their capacity. They explained that despite their lack of experience in caring for circumcised women, they were not anxious or nervous about working around FGM/C. They argued that rather than being confronted, they felt comfortable. This was in terms of providing general support of listening, guidance, and helping affected women and girls to find the right avenues in optimising their bodily functioning, such as referral to the hospital.

## 4. Discussion

The study was able to capture an in-depth understanding of the experiences of primary healthcare professionals regarding their general knowledge, experience, support, and care for girls and women in Australia living with FGM/C. From the results, participants acknowledged having basic knowledge and information about FGM/C, which was similar to findings of previous studies focused on determining the level of knowledge about FGM/C among healthcare providers [28,38,39]. Furthermore, the results may be due to the heterogeneity of the participants. The participants described their knowledge about FGM/C as very limited and that they had little information about the practice. This was consistent with the Belgian study that revealed gaps in knowledge about FGM/C among Flemish gynaecologists [40] and Australian midwives [41]. 

This lack of knowledge identified by the majority of the study participants can be linked to a lack of education and training. Andro et al. [42] indicated that FGM/C education is important. In high-prevalence FGM/C settings, Relph and colleagues [39] discovered that less than 25% of practitioners had formal FGM/C training. The study also found that healthcare professionals had significant gaps in their knowledge about how to manage women with FGM/C, highlighting the need for comprehensive FGM/C training [39,43]. The study cohort indicated that without adequate educational training, addressing the psychosocial issues of these women (presenting for care) was almost impossible. This led them to taking matters into their own hands. For example, the study participants highlighted that underwent self-study journeys, such as reading books and watching documentaries, to help them become more proficient in care for women and girls who have experienced FGM/C. Nevertheless, this effort, while noble, still provided patchy knowledge and competency that may affect their confidence and attitude in providing quality care for the target population [44]. The study by Marea et al. [44] advised that future training should include FGM/C instructional content (history and context, identification of the four types, documentation, complications, and management of women living with FGM/C) to increase provider confidence in their ability to offer clinical care for women with FGM/C. Training should include opportunities for simulation, reflection, and participation in order to practise applying the information and communication skills, as well as improve the understanding of how attitudes may affect the quality of care provided to persons affected by FGM/C [45].

This study has bridged the previously identified lack of doctors’ voices in empirical research among primary healthcare providers in relation to knowledge about FGM/C [29]. Even though there was only one GP in the cohort, this can be considered a small win for further exploration. Perhaps the underrepresentation of medical doctors could be due to the busy nature of their work. However, it can also be argued that the health issues of women and girls living with FGM/C in the community is part of their business and they are integral in the provision of continuity of care for affected women. Thus, interest in research that can inform their capacity to provide optimal care to women and girls with FGM/C needs to be demonstrated.

It is significant to reiterate that Australian health systems in recent years have put measures in place to breach the stated educational gaps [22,23,24]. For example, the NSW education program FGM booklet and National education toolkit for FGM/C awareness (NETFA) are aimed at all healthcare professionals who may come into contact with women, girls, and families affected by FGM/C, providing a general overview of the practice, policies, laws, and who is most impacted in the community. While this training is important for the knowledge and skill development of healthcare practitioners, the majority of the study cohort indicated that they had no encounters or experience in using their developed skills and knowledge to care and support the target group, which made their knowledge and skills stale and dated. Some of this training is not compulsory learning criteria for healthcare providers and the content may only be relevant to a particular state that is a gap for the target population and those impacted by the practice. Studies over time among healthcare professionals have evidently shown that encounters and experiences with women and girls affected by FGM/C occur during the training process of healthcare practitioners [46,47,48]. Doctors, nurses, and midwives in the United States [46], healthcare professionals in Norway [47], and midwives in Sweden [48] all reported receiving no formal training in the management of these affected women or girls, but they were provided with opportunities to learn “on the job” or from more senior or more experienced colleagues. This was similar to our study findings, where some participants (n = 5) who had identified with the majority as not having any comprehensive knowledge on FGM/C and management had more experience from encounters with women or girls living with FGM/C. 

Cultural competency and safety for this target group in healthcare settings could be affected because education, training, and encounters for primary healthcare providers are limited, as established earlier [27]. This could mean that healthcare practitioners’ attitudes towards this population of interest may be incompetent [44]. The study findings indicated that the majority of women living with FGM/C who accessed health services lacked a “willingness to disclose information about their FGM/C”. This could be attributed to distrust felt towards the practitioner providing the service [49] or the health system. Some participants highlighted the reason for hesitance was due to “lack of trust in healthcare providers,” “fear of mandatory reporting,” and “thought of helplessness”. According to the study by Ogunsiji [41], when a healthcare worker encounters a patient who has undergone FGM/C, they may experience a strong affective response such as anger towards the procedure of FGM/C or bias towards the women who have undergone it. This attitude may obstruct the patient-provider interaction [50]. 

Evidence from the studies by Brown et al. [51] and Hamid et al. [52] suggested that some women with FGM/C prioritise minimal interventions. This may be perceived by providers (who may not have actively listened to the patient’s perspective or priorities) as antagonistic or ignorant. The current study cohort indicated similar feelings highlighted by these studies. Nonetheless, the study participants spoke about the importance of building rapport, trust, and genuine interest and desire to engage with women and girls with FGM/C. To buttress this point, Dahlberg and Aune [53] in their study of interpersonal relationships and continuity of care stated that positive engagement, rapport, and trust-building increases interpersonal communication and confidence in women. It also fosters collaboration between a woman and her provider and facilitates women’s engagement in the process of care design and delivery [53]. We suggest that any interventions aimed at promoting therapeutic relationships with women living with FGM/C need to incorporate training sessions on rapport and trust-building. This is crucial in enhancing optimal health outcomes. 

## 5. Recommendation

This study has clearly established the lack of formal education, training, and experience of Australian primary healthcare workers in providing support and competent care for women and girls living with FGM/C in Australia. While there are existing resources available, as stated earlier, more needs to be done to breach the existing identified gaps. The role of training and tertiary institutions in promoting knowledge of FGM/C among healthcare providers requires urgent attention. We suggest that topics on FGM/C need to be embedded in the pre-tertiary, tertiary, and post-tertiary education of women’s health nurses, counsellors, doctors, social workers, and other healthcare providers in community settings. According to Molina-Gallego et al. [54], this could be a great solution. In addition, increasing awareness among healthcare workers, more multidisciplinary teamwork, and higher quality training could also help to improve the care and management of women affected by FGM/C. A similar situation was observed by Kaplan et al. [55] who discovered an increase in the number of FGM/C cases identified after carrying out various educational initiatives in healthcare centres. This point was corroborated by Turkmani et al. [26], who suggested that a nationwide strategy for training and education of Australian healthcare providers may be the critical factor in ensuring that the highest possible quality of treatment is delivered. Furthermore, FGM/C is a global health issue that requires global attention. Regardless of the participants being healthcare providers or part of the global community, there is need for awareness to advocate against the practice.

Marea et al. [44] and Atkinson et al. [45] provided comprehensive summaries on what can be embedded in the training. They mentioned that in order to practise applying the acquired knowledge and communication skills on those affected by FGM/C, training should include opportunities for simulation, reflection, and engagement. Training should also raise awareness about how attitudes may affect the quality of care delivered to those affected by FGM/C. Communication skills training must involve the employment of interpreters, including the advantages and risks of doing so, as well as the impact on provider-client rapport, trust, and ethical considerations. Instruction on reflective practice should be included in all training for healthcare workers who work with women who have undergone FGM/C or who work with any marginalised community.

The significantly low representation of medical doctors’ voices in this study requires urgent attention, given their important and leading role in the health system. We suggest targeted research among medical professionals to gain deeper insight into their understanding of this global health issue and identify areas for capacity building towards optimal care of affected women.

Further, the research by Njue et al. [17] recommended the need for training on referral pathways, which was indicated as an issue for primary healthcare professionals in this study. Njue and colleagues [17] suggested FGM/C prevention and care initiatives for trainees and practising healthcare providers. Community involvement in high-burden areas/populations is recommended to ensure more people use and take advantage of the referral service. This can be performed through infrastructural support, such as clinical management tools, job aids, posters, referral algorithms, and electronic patient records with drop-down menus for referral sites for health complications related to FGM/C, in order to support the providers’ efforts.

## 6. Limitation

This study had some drawbacks. First, the restricted number of medical professionals in the sample may limit the generalisability of the results for clinical healthcare practitioners. In addition, the sample may be constrained by a selection bias due to the participants’ keen interest in the research issue. Sample heterogeneity can be considered a limitation of this study; however, this provided a range of perspectives from primary healthcare settings where there is a paucity of research. Further, the collected data consisted of self-reports from primary healthcare providers. Self-reporting may not reflect actual levels of knowledge or experience obtained when interacting with the target group [56].

## 7. Conclusions

The majority of the participants in the study had limited knowledge of FGM/C, despite the fact that they were more likely to encounter affected women and girls. The depth of knowledge was patchy, and engaging in sustained conversation with the women was challenging. Meanwhile, this is an increasing global health issue that should be prioritised in western countries. This is particularly true in Australia and many other countries with increasing populations of women and girls from countries where FGM/C is prevalent. Hence, the healthcare system at the organisational, professional, and individual levels needs to be provided with comprehensive education and training. This is germane to promoting proficient and competent healthcare that will positively impact the FGM/C affected population seeking healthcare services. 

### What Does This Paper Contribute to the Wider Global Clinical Community?

This paper brings to the fore that healthcare providers in Australia need to be more knowledgeable in the care, management, and support of women/girls living with FGM/C.The importance of increasing FGM/C awareness among relevant healthcare workers in the primary healthcare setting, particularly in the Australian context.There is a dire need for more training and capacity building on health matters influenced by FGM/C.This paper also highlights the need for infrastructure and resources for providing culturally sensitive care to this target population in order to increase trustworthiness.

## Figures and Tables

**Table 1 healthcare-11-00702-t001:** Major themes and subthemes.

S/N	Major Themes
1.	Exploring knowledge of FGM and training needs. Sub-themes: • Having a lack or patchy knowledge of FGM/C. • Having a negative attitude towards FGM/C. • Need for training.
2.	Understanding participants’ experience in caring for women living with FGM/C.
3.	Mapping the best practice in working with women.

## Data Availability

The data are not publicly available; however, this can be provided on request.

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
