# Peer review of "Experiences of Primary Healthcare Workers in Australia towards Women and Girls Living with Female Genital Mutilation/Cutting (FGM/C): A Qualitative Study"

_healthcare, 2023, doi:10.3390/healthcare11050702_

Round 1
Reviewer 1 Report
Thank you for the opportunity to review this important work on female genital mutilation. Overall, the paper is well written. In the Introduction section, the author(s) have done an excellent job providing a comprehensive overview of the issue of FGC/M globally and in Australia, including its legal and policy provisions, efforts to promote cultural competence, and ongoing initiatives to address the needs of affected women and girls. The discussion of the extant literature on FGC/M knowledge and awareness amongst healthcare provider in Australia also provide helpful context.
The authors use FGC/FGM interchangeably. To provide additional contextual information, the authors should include a discussion on the distinctions between those two terms and provide a rationale for using both. For instance, in my work, I choose to use the term “female circumcision” in lieu of “female genital mutilation” to describe the cultural practice without judging those who practice it. This is a controversial topic and it needs to be addressed in the paper.
Aim and Research Question: The author should specify that the focus of the study was Australian primary health care providers.
Recruitment and Sample: It is not clear who the target population for the study was. The inclusion criteria need to be more specific. How does the research define a “healthcare provider”? The aim of the study was to explore the experiences of primary care providers. However, a legal assistant and administrative staff were recruited for the study. Was study eligibility open to only health and social service professionals in a primary care setting? Or could any worker (for instance, a sanitation worker) in a primary care setting enroll in the study? If any worker could enroll in the study, then is the term “healthcare provider” appropriate?
The study findings were well-presented.
Discussion: My greatest challenge with this paper is that a stronger case needs to be made for the significance of the study. The author(s) identify the inclusion of one GP as a novelty in the study. That claim is arguable given that there was only one GP. Also, sample heterogeneity is a significant limitation in this study. Generally, FGC knowledge and experience will vary based on training, role, and the nature of the individual’s interactions with patients.
Line 505: The author(s) should expand on what measures are being implemented to address FGC knowledge gaps in healthcare providers.
Author Response
Dear Reviewer,
The team would like to thank you for your taking the time to review this manuscript and for providing feedbacks to items to improve upon.
The authors have reviewed your feedback and provided responses below:
The authors use FGC/FGM interchangeably. To provide additional contextual information, the authors should include a discussion on the distinctions between those two terms and provide a rationale for using both. For instance, in my work, I choose to use the term “female circumcision” in lieu of “female genital mutilation” to describe the cultural practice without judging those who practice it. This is a controversial topic and it needs to be addressed in the paper. This has been addressed in the Australian context using credible resources which is noted in item 1.3
Aim and Research Question: The author should specify that the focus of the study was Australian primary health care providers. This has been addressed in the conclusion of the introduction on line 121.
Recruitment and Sample: It is not clear who the target population for the study was. The inclusion criteria need to be more specific. How does the research define a “healthcare provider”? The aim of the study was to explore the experiences of primary care providers. However, a legal assistant and administrative staff were recruited for the study. Was study eligibility open to only health and social service professionals in a primary care setting? Or could any worker (for instance, a sanitation worker) in a primary care setting enroll in the study? If any worker could enroll in the study, then is the term “healthcare provider” appropriate? The authors have provided a justification in 2.3 recruitment and sample section with further touch on the point in section 3.
Discussion: My greatest challenge with this paper is that a stronger case needs to be made for the significance of the study. The author(s) identify the inclusion of one GP as a novelty in the study. That claim is arguable given that there was only one GP - Authors have provided justification on this point which be found in lines 516 -520 provided on line.
Also, sample heterogeneity is a significant limitation in this study - Authors have noted this in the limitation section in line 623.
Generally, FGC knowledge and experience will vary based on training, role, and the nature of the individual’s interactions with patients - The authors argues that in a country such as Australia where there is high percentage of migrants and refugees from Sub-Sharan Africa, is it not imperative that a standard training should be prioritised for the reproductive and sexual wellbeing for this population? from the data gathered the authors know that is where is the significance of this study lies to highlight the gaps in the experience and training of the health workforce and recommendations on how it can be better.
The author(s) should expand on what measures are being implemented to address FGC knowledge gaps in healthcare providers - Authors have provided the information recommended from line 525 - 529 & 533 -535

Author Response
Dear Reviewer,
The team would like to thank you for taking the time to review and for providing feedback on our manuscript.
The authors have made the recommended adjustment to different areas where it was noted.
Title and abstract
The title to include the setting (Australia) of the study. This has now been reflected in the title.
On lines 18 and 19 in the abstract there is a repetition of themes identified. I suggest it stays written only in line 19. This has been revised in line 23.
Introduction
On page 2, in the paragraph starting on line 53, maybe you could mention the ages when FMG usually takes place and describe more thoroughly possible immediate and long term effects. The age range has been added in line 57. The authors have decided not to go beyond with the information provided on the immediate and long term effects as the authors believe the information provided is sufficient for the reader to understand the impact of the practice.
There is no need for a section 1.2 on page 3, as the aim is already explicit on lines 119-21. This has been taken out as recommended.
Materials and Methods
What was the time period for the data collection ? This could be included on section 2.4 in page 4. The timeline has now been added to section 2.4 on data collection.
Perhaps the interview schedule could be in annex and not in main text. The team decided, that the interview schedule gives clarity, transparency of the process to the reader in section 2.4
Conclusion
To allow for easier reading and better focus, the conclusion section could be shortened and without subsections. It is repetitive for the SDG comments, for example. The conclusion has been shortened to provide clarity to the reader.

Round 2
Reviewer 1 Report
All concerns raised by this reviewer have been addressed appropriately.